# Empathy and COVID-19: Study in Professionals and Students of the Social Health Field in Ecuador

**DOI:** 10.3390/ijerph18010338

**Published:** 2021-01-05

**Authors:** Ana J. Cañas-Lerma, M. Elena Cuartero-Castañer, Guido Mascialino, Paula Hidalgo-Andrade

**Affiliations:** 1Philosophy and Social Work Department, Universitat de les Illes Balears, Crta. Valldemossa Km 7.5., 07122 Palma, Spain; ana.lerma@uib.es; 2School of Psychology, Universidad de Las Américas, Av. de los Granados E-12-41 y Colimes esq., Quito EC170125, Ecuador; guido.mascialino@udla.edu.ec (G.M.); paula.hidalgo@udla.edu.ec (P.H.-A.)

**Keywords:** empathy, COVID-19, health personnel, university students, Ecuador

## Abstract

Empathy plays a fundamental role in health related occupations. In this study, we analysed empathy levels in professionals (117) and students (170) from various healthcare fields in Ecuador during the COVID-19 pandemic. The Interpersonal Reactivity Index was used in an online survey. The results show high levels of empathy in both groups, influenced by age and gender. The students presented higher levels of personal distress, and their age was negatively correlated to empathy. Additionally, professionals working in physical health scored higher levels of personal distress compared to those in the field of emotional health. COVID-19 has placed social health systems in great stress. Despite this, the personal capacities for empathy of both students and health professionals have not been diminished.

## 1. Introduction

Although there is controversy over its definition, empathy refers to the variety of reactions of one person to another’s observed experiences, and it has four components: perspective taking, empathic concern, personal distress, and fantasy [1]. This study uses the multidimensional approach to empathy and incorporates the union of the cognitive and affective dimensions. Affective empathy is the ability to experience users’ emotions and perspectives, while cognitive empathy is the ability to identify and understand, from one’s perspective, the emotions of others. The cognitive dimension comprises subconstructs, such as perspective taking and fantasy, while the emotional dimension contains empathic concern and personal distress as sub-components.

Empathy has been highly studied, and it has been associated with spirituality [2] and prosocial behaviour, a voluntary behaviour intended to benefit others. A study identified that empathy and moral identity were the potential mechanisms underlying the association between intelligence and prosocial behavior [3]. Research also suggests empathy can be nourished and developed [4].

There is a growing body of work focusing on empathy as a multidimensional construct in the health professions. Recent research [5] conducted with Chinese medical students showed that women displayed higher empathy levels than their male peers. This same study noted that empathy declined as students progressed in their studies. Schoenfeld-Tacher, Shaw, Meyer-Parsons, and Kogan, [6] used the Interpersonal Reactivity Index (IRI) in a sample of veterinarians. Their findings revealed that with increasing age and therefore, expertise, fantasy and personal distress decreased. These results are consistent with research carried out by Bratek, Bulska, Bonk, Seweryn, and Krysta [7] on medical students and physicians, which found a negative correlation between age and fantasy as well as personal distress. However, in this same study, a positive correlation was found between age and perspective taking.

Moreover, empathy plays a vital role for people in health related professions [8]. For instance, Hunt [9] found decreased compassion fatigue in cancer healthcare professionals with higher levels of empathy while Mc Farland [10] noted greater resilience in an oncology setting associated with empathy. Besides, Shanafelt [11] found increased personal wellbeing in medical residents with higher levels of empathy. Empirical research shows that it is essential to work effectively with others in suffering, contribute to the development of a strong therapeutic relationship, and correlate with better clinical outcomes [4,12]. Furthermore, developing empathy is necessary to become professionally competent and have higher professional fulfilment [13].

Given what we already know about empathy and the consideration that empathic communication and response during a healthcare crisis could lead to better support [14], this report aims to explore it in students and professionals in health related occupations in Ecuador during the COVID-19 pandemic. This research is of particular interest, because the global health crisis outbreak could and likely did generate remarkable emotional changes in healthcare providers who usually do not have specific training in empathy or dealing with their own emotions. In addition, this study is relevant because of the abrupt changes to the way people relate to each other and perform their tasks brought about by the significant measures imposed in Ecuador to protect people’s physical health, which included a national lockdown, curfews, and the cancellation of all non-essential in-person activities for over five months.

## 2. Method

### 2.1. Participants

Participants were recruited in Quito, Ecuador, through snowball sampling for this cross-sectional quantitative study. Inclusion criteria for the study were: being 18 years old or older, living in Quito, and being university students or professionals in either physical (medicine, nursing, and nursing assistants), emotional (social work and psychology), or rehabilitation (physiotherapy, occupational therapy, and others) health related fields. The sample consisted of 287 subjects: 170 students from eight universities and 117 professionals from nine health centres in the city with an average of 14.35 (SD = 12.19) years in their job positions. The students did not receive any specific training on empathy beyond the content of their subjects. Additionally, 92% of the professionals surveyed acknowledge having carried out specific health training in the last 5 years, but not in prosocial content.

### 2.2. Instruments

Data collection took place through an online questionnaire that included sociodemographic variables, such as age, gender, marital status, and professional/occupation speciality. In the case of professionals, they were asked about the number of years of professional experience, the number of patients they attended weekly, hours of work per day, percentage of cases of high emotional burden, and self-care practices.

The Interpersonal Reactivity Index questionnaire (IRI) [1], in its Spanish translation [15], was also administered to measure the total empathy levels of the participants, as well as its four sub-dimensions (fantasy, empathic concern, perspective taking, and personal distress). Fantasy is the tendency to identify with characters in movies, novels, plays, or fictional situations. In the questionnaire, some of the items that measure this subdimension are: “I daydream and fantasize, with some regularity, about things that might happen to me” or “I really get involved with the feelings of the characters in a novel.” Empathic concern involves feelings of compassion, concern, and affection for the discomfort of others (e.g., “I often have tender, concerned feelings for people less fortunate than me” and “I am often quite touched by things that I see happen”). Perspective taking is the tendency to adopt the psychological point of view of others spontaneously. Example statements of perspective taking from the IRI include “I try to look at everybody’s side of a disagreement before I make a decision” and “I sometimes try to understand my friends better by imagining how things look from their perspective.” Finally, personal distress is the emotional reactions, discomfort, and anxiety to others’ negative experiences. Items such as “being in a tense emotional situation scares me” or “when I see someone who badly needs help in an emergency, I go to pieces” represent personal distress in the scale.

The IRI has been widely used in the health field’s scientific area since its creation in 1983 [1]. It is currently one of the most frequently used questionnaires to measure empathy levels in health students and professionals. This instrument has shown satisfactory psychometric properties [16]. The Cronbach’s Alphas in this study were between 0.70 and 0.78 in each of the dimensions.

### 2.3. Procedure

Health centers and universities were contacted to participate in the study. After approval by the Ethics Committee of the Universitat de les Illes Balears (Exp.num143CER20) and Universidad de Las Américas (cod. 20200414), an email explaining the study including a link to the questionnaire, developed through Google forms, was sent to the health centres and their members, as well as published on their websites and social networks (Facebook and Twitter). Data collection took place from April to July 2020. During this period, Quito was in lockdown due to COVID-19.

### 2.4. Analyses

All statistical analyses were carried out using SPSS version 20. Reliability and normality tests were performed, followed by descriptive and frequency analysis of the sample. Pearson’s correlations, student *t*-tests, and ANOVAs were conducted for univariate analysis. A linear regression model was also performed to consider the effect of multiple variables on empathy levels.

## 3. Results

The empathy scale had a Cronbach’s Alpha of 0.762; its subscales also showed right internal consistency (fantasy: 0.795; empathy concern: 0.647; perspective taking; 0.700, and personal distress: 0.717). The Kolmogorov–Smirnov test was greater than 0.05; thus, all the performed tests were parametric. The main sociodemographic characteristics of the sample are shown in Table 1.

There were no significant differences in gender between the groups; there was a greater female to male ratio in both cases. However, professionals were significantly older (F(1282) = 244.65, *p* < 0.01). The average age of the full sample was 30.80 years (SD = 12.03), that of the professionals was 40.65 years (SD = 12.55), while that of the students was 23.99 (SD = 4.77). Professionals were also more likely to be married (X^2^ (3, N = 286) = 98.31, *p* < 0.001) than students and had significantly more representation in the physical health field versus the emotional health field in the students (X^2^ (2, N = 285) = 56.98, *p* < 0.001). Lastly, more than 70% of the students were about to finish their careers and had already carried out internships in work centres. In the case of professionals, they had an average of 8.07 years of experience at their workplaces (SD = 7.9) and attended an average of 74.97 patients weekly (SD = 46.2). They also indicated that 44% of the number of weekly cases were of high emotional distress.

Table 2 shows the average levels of empathy, its dimensions for professionals and students, and the combined sample. The *t*-test results between professionals and students show statistically significant differences between professionals and students in all the categories. Students show higher levels of empathy, fantasy, empathic concern and personal distress, while professionals have higher levels of perspective taking.

When considering the sample as a whole (Table 3), effects of gender were noted on the total level of empathy, fantasy, empathic concern, and personal distress, in which women had higher scores than men in all the analyses. The marital status differences were also noted in total empathy, with singles having the highest scores over the other groups. However, post hoc analyses did not reveal significant differences between the groups. Singles also had significantly higher levels of personal distress compared to divorced participants after post hoc comparisons. Empathic concern, perspective taking, and fantasy levels were also significantly different between the groups. Post hoc analyses revealed that singles had higher levels of empathic concern than common law married participants, lower levels of perspective taking than married people, and higher fantasy levels than divorced participants. Age was significant and negatively correlated with all empathy variables (total empathy *r*(284) = −0.203, *p* = 0.001; fantasy *r*(284) = −0.300, *p* < 0.001; empathic concern *r*(284) = −0.205, *p* = 0.001; perspective taking *r*(284) = −0.241, *p* < 0.001; personal distress *r*(284) = −0.214, *p* < 0.001). Differences were also noted according to health field, but only for perspective taking and empathic concern. Post hoc analyses revealed participants in the emotional health field had higher empathic concern levels than all other groups. Still, no post hoc significant differences were noted on perspective taking.

According to sociodemographic and work variables within each subsample, different patterns emerged when analyzing differences in empathy; age was associated with students’ empathy levels but not professionals. Besides, gender was associated with most empathy variables in students, but only one in professionals. Differences in health related fields were significant only in one variable in the professional’s group and none in students.

The Pearson’s correlation results show that age was significantly and negatively associated with total empathy levels in students. As the age of the students increased, the total levels of empathy decreased (*r*(168) = −0.245, *p* = 0.001). Similarly, the dimensions of fantasy (*r*(168) = −0.295, *p* < 0.001), empathic concern (*r*(168) = −0.175, *p* = 0.023), and personal distress (*r*(168) = −0.167, *p* = 0.030) also decreased as age advanced in students. In addition, empathy levels were not associated with the number of patients seen weekly in professionals.

As seen in Table 4, when comparing total empathy by gender, only perspective taking was significantly different in the sample of professionals. On the other hand, in the student sample, women had higher mean levels of empathy, perspective taking, empathic concern, and fantasy than men. Effects of the health field, whether professions were related to physical, mental, or rehabilitation health, on total empathy were noted only in the professionals’ sample. Those working in the physical health field had significantly higher personal distress values than those working in emotional health.

Lastly, a linear regression model was performed to see if occupation (professional vs student), age, cohabitation status (living with a romantic partner or not), health field, or gender variables could predict empathy and its dimensions. For each model, we first included all the above-mentioned variables and then removed the non-significant ones. For total empathy, we removed health field (β = −0.054, *p* = 0.965), occupation (β = 0.104, *p* = 0.200), and cohabitation status (β = 0.117, *p* = 0.075). The model with age (β = −0.174, *p* = 0.003) and gender (β = 0.232, *p* = 0.000) was significant (F(2, 280) = 14.456, *p* < 0.000, *R*^2^*_adjusted_* = 0.087). For fantasy, the non-significant variables were health field (B = −0.003, *p* = 0.375), occupation (β = 0.077, *p* = 0.963), and cohabitation status (β = 0.104, *p* = 0.113). Again, the model with age (β = −0.283, *p* = 0.000) and gender (β = 0.136, *p* = 0.017) was significant (F(2, 280) = 16.831, *p* < 0.000, *R*^2^*_adjusted_* = 0.101). In the case of empathic concern, the non-significant variables were age (β = 0.013 *p* = 0.869) and cohabitation status (β = −0.030, *p* = 0.646). The model with the remaining variables, occupation (β = 0.216, *p* = 0.000), gender (β = 0.178, *p* = 0.002), and health field (β = 0.129, *p* = 0.031) was significant (F(3, 280) = 13.593, *p* < 0.000, *R*^2^*_adjusted_* = 0.118). Given that occupation was a significant factor in this model, we performed the regression for the students and professionals’ subsamples separately. Only the model for students was significant. Gender (β = 0.308, *p* = 0.000) was the only variable with an effect (F(1, 167) = 17.471, *p* < 0.000, *R*^2^*_adjusted_* = 0.089). For perspective taking, the non-significant variables removed from the model were age (β = −0.021, *p* = 0.788) and health field (β = −0.071, *p* = 0.235). The remaining variables, gender (β = 0.183, *p* = 0.001), occupation (β = −0.280, *p* = 0.000), and cohabitation status (β = 0.124, *p* = 0.045), were significant for the model (F(3, 281) = 15.249, *p* < 0.000, *R*^2^*_adjusted_* = 0.131). As with empathic concern, the model was run separately for students and professionals, but they were not significant. Lastly, in the regression model for personal distress, the removed variables were health field (β = 0.001, *p* = 0.991), age (β = −0.093 *p* = 0.260), gender (β = 069, *p* = 0.239), and cohabitation status (β = 0.086, *p* = 0.197). Only occupation (β = 0.243, *p* = 0.000) remained in the model (F(1, 285) = 17.825, *p* < 0.000, *R*^2^*_adjusted_* = 0.056).

## 4. Discussion

The impact of the COVID-19 health and social crisis in Ecuador, during which the present data was collected, seems to be reflected in the professionals’ responses, who reported that almost half of the weekly cases they attended presented a high emotional load. Despite this, the high levels of empathy obtained were in line with those collected from other contexts and methods [17]. The fact that the professionals who attend to physical health care needs have presented the highest level of personal anguish is consistent with other research [18] and is understandable since they are the ones in direct contact with death overflowing in the health system.

In this study, the students show greater distress and concern than experienced professionals. The instability related to the current global socio-health panorama also disrupted university educational environments. Students in health related fields have had to substantially change their study routines and access to learning due to the imposed confinement measures. Despite all the changes, a high level of total empathy was observed in students. This could be explained by having access to information about the situation of those affected by the disease and being privy to the difficulties of the social health professionals who were taking care of them and by whom they were being trained [19]. On the other hand, the high levels of personal distress could be explained by their efforts to adapt to the new reality, both academically and in their future profession. Likewise, many of them have carried out their clinical practices. As Taylor, Thomas-Gregory, and Hofmeyer [20] point out, they may have experienced a deficit in professional supervision, a lack of personal protective equipment, and fear of personal contagion or exposure to it for their families.

In this study, professionals have greater perspective taking, probably due to professional experience. One of the most relevant results of this research is the relationship between perspective taking and age. This confirms that the greater the professional experience in the health field, the greater the person’s ability to understand others while they taking a healthy distance. These data are consistent with the findings of Bratek, et al. [7]. Interestingly Richards, Petty, and Zelenski [21] found results contrary to the present study. Their sample of new professionals and students in genetic counselling found that first year students had greater perspective taking than sophomores or graduates. However, this study does not take into account professionals with more years of professional practice. If more veteran professionals had been included, perhaps, results more in line with those proposed in the present study could be found.

As Harrison and Westwood [22] point out, emotional distancing allows more effective help to patients. Health professionals mark the necessary limits for the correct development of their professional empathy. It should be noted that too much distance can lead to inattention to the patient and dehumanisation of care. On the other hand, excessive bonding with the patient can lead to resonances and an overexpansion that makes it difficult to take perspective. The regulation of said emotional distance can be a preventive mechanism against burnout and secondary traumatic stress derived from professional practice.

Furthermore, the present research shows that age and gender have an impact on total empathy levels. As in previous studies [5,23,24], female students have higher empathy levels than male students. In contrast to what was presented in Extremera Pacheco and Fernández Berrocal’s research [23], the students decreased their level of empathy as their age increased in the present study. This data should be the object of further research since it may be derived from the exceptional situation in which the participants responded since the previous literature are not in line with these results.

COVID-19 has raised many moral dilemmas [18]. However, the high levels of empathy obtained in this study likely reflect the determination of healthcare providers fighting the pandemic and accompanying those who suffer, despite technical and material difficulties or shortages. Therefore, attention to empathy, as well as personal and organisational self-care should not be forgotten [20] since greater emotional management is positively related to greater empathic concern, better perspective for students and professionals, and less personal distress for professionals [23].

This research is not without limitations. We do not have at our disposal measurements before the onset of COVID-19, and as a result, the pandemic’s impact on the levels of empathy of professionals and students cannot be ascertained. Having previous measurements would have made it possible to determine the fluctuations in levels concerning the stress and tension experienced in the social health area. Future studies should measure the changes in empathy as the pandemic advances. Additionally, future research should include a bigger sample size. Although, considering the extraordinary limitations and difficulties of confinement, this study has representative sample results and opens the door to future studies on the subject.

Empathy is a double-edged sword. It is essential to establish an adequate helping relationship, but at the same time, it may induce an emotional overinvolvement with those who suffer. This lack of empathy balance can lead to negative professional practice consequences, such as burnout, depression, or post-traumatic stress. Empathy scores can serve as traffic lights to warn of occupational hazards. The high levels of empathy obtained in this study should alert healthcare managers to increase their staff’s care to avoid the abovementioned consequences.

The COVID-19 pandemic has shown the importance that health professionals have in fostering community wellbeing. Understandably, the lion’s share of the responsibility has fallen onto them to provide ongoing support to address this global emergency. The number of stressors to which they have been subjected may have increased their burnout levels, secondary traumatic stress, loss of quality in professional life, and satisfaction with life [25]. Studies such as this one help to support professionals in their practice by allowing us to explore how the current global crisis has impacted them. Based on these results, responses aimed at improving their wellbeing can be proposed. This study is a first step that focuses on empathy as a skill, in which the optimal relationship between professionals and patients is sustained. Ultimately, empathy is an essential skill for health professionals as it promotes clinical competence, patients and professionals’ wellbeing, and trust and emotional intelligence [26]. Work from a humanistic model, in which both the professional and the patient are essential, contribute to building a more efficient health model in these critical moments.

## Figures and Tables

**Table 1 ijerph-18-00338-t001:** Sociodemographic characteristics of participants.

	Professionals (*n* = 117)		Students (*n* = 170)		Full sample (N = 287)	
	*n*	%	*n*	%	*n*	%
Gender						
Female	74	63.2	123	72.8	197	68.9
Male	43	36.8	46	27.2	89	31.1
Marital status						
Single	45	38.8	156	91.8	201	70.3
Married	42	36.2	6	3.5	48	16.8
Partnered	7	6.0	5	2.9	12	4.2
Divorced	22	19.0	3	1.8	25	8.7
Health field						
Physical	74	63.2	34	20.2	108	37.9
Emotional	23	19.7	93	55.4	116	40.7
Rehabilitation	20	17.1	41	24.4	61	21.4

**Table 2 ijerph-18-00338-t002:** *t*-test results comparing empathy between professionals and students.

	Professionals (*n* = 117)		Students (*n* = 170)		*t*(285)	*p*	Hedges’ *g*
	M	SD	M	SD			
Empathy	88.35	13.08	92.94	11.63	−3.12	0.002 **	0.37
Fantasy	21.15	5.61	23.81	5.71	−3.89	<0.001 **	0.47
Perspective Taking	27.91	4.93	24.86	4.45	5.45	<0.001 **	0.66
Empathic Concern	26.03	4.17	28.74	4.82	−4.93	<0.001 **	0.59
Personal Distress	13.26	4.80	15.54	4.27	−4.22	<0.001 **	0.51

** *p* < 0.01, two-tailed.

**Table 3 ijerph-18-00338-t003:** ANOVAs of empathy levels by gender, marital status, and health field for the entire sample.

		df	MS	F	*p*
Empathy	Gender	1	2748.62	18.91	<0.001 **
	Marital	3	447.991	2.961	0.033 *
	Field	2	158.703	1.027	0.359
Fantasy	Gender	1	277.725	8.417	0.004 **
	Marital	3	107.204	3.244	0.022 *
	Field	2	20.203	0.599	0.55
Perspective Taking	Gender	1	124.8	5.321	0.022 *
	Marital	3	145.026	6.42	<0.001 **
	Field	2	86.74	3.721	0.025 *
Empathic Concern	Gender	1	309.311	14.326	<0.001 **
	Marital	3	89.452	4.079	0.007 *
	Field	2	190.9	8.904	<0.001 **
Personal Distress	Gender	1	49.05	2.297	0.131
	Marital	3	57.753	2.803	0.04 *
	Field	2	40.686	1.914	0.149

* *p* < 0.05, ** *p* < 0.01, two-tailed.

**Table 4 ijerph-18-00338-t004:** ANOVAs of empathy levels by gender, marital status, and health-field by sample subgroup.

		Professionals				Students			
		df	MS	F	*p*	df	MS	F	*p*
Empathy	Gender	1	575.149	3.43	0.067	1	1950.728	15.655	<0.001 **
	Marital	3	256.29	1.524	0.212	3	93.479	0.687	0.561
	Field	2	142.366	0.829	0.439	2	61.886	0.454	0.636
Fantasy	Gender	1	36.752	1.169	0.282	1	198.269	6.227	0.014 *
	Marital	3	21.511	0.678	0.567	3	21.279	0.647	0.586
	Field	2	4.46	0.14	0.87	2	8.4	0.256	0.774
Perspective Taking	Gender	1	99.261	4.199	0.043 *	1	93.116	4.833	0.029 *
	Marital	3	47.149	1.984	0.12	3	7.451	0.372	0.773
	Field	2	59.629	2.519	0.085	2	57.706	3.028	0.051
Empathic Concern	Gender	1	2.638	0.15	0.699	1	371.204	17.471	<0.001 **
	Marital	3	14.436	0.818	0.486	3	59.738	2.648	0.051
	Field	2	27.316	1.585	0.21	2	57.955	2.519	0.084
Personal Distress	Gender	1	40.104	1.751	0.188	1	1.369	0.074	0.786
	Marital	3	10.924	0.494	0.687	3	4.301	0.233	0.873
	Field	2	73.618	3.321	0.04	2	24.756	1.368	0.257

** p* < 0.05, ** *p* < 0.01, two-tailed.

## Data Availability

The data presented in this study are available on request from the corresponding author. The data are not publicly available due to ongoing longitudinal analysis.

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
