# Peer review of "Empathy and COVID-19: Study in Professionals and Students of the Social Health Field in Ecuador"

_ijerph, 2021, doi:10.3390/ijerph18010338_

Round 1

Reviewer 1 Report

The author examine the determinants of "empathy" among health professionals and students in Quito during the Covid-19 lockdown.

Although it is still early in publication stream of many journal, this is undoubtedly a unique study. A bit of a snapshot of social turmoil generated by the pandemic. Under normal condition, the sample size would be relatively small. However, given the situation, the sample size is acceptable. AS there are no page limits here, a discussion of the some of the obstacles conducting this study would be informative.

Three comments:

1) I think a little editing of the discussion section would make it easier to read and understand. Specificially, shorter paragraphs focusing on one point.

2) I don't think too many of us should be surprised by the findings. However, I think the importance of this paper would be raised by a brief discussion of your findings in relationship to other stressors. [Or tell us more about why this study was undertaken.]

3) A larger number of subjects would have been better (always better). However in the age of Covid, this is very good.

Some comments in the pdf may well be considered minor.

Author Response

Dear reviewer 1,

We want to thank you, wholeheartedly for your feedback. We think that your comments have helped us to fine-tune and improve the standard of the paper. In the following points, we reply to your constructive remarks.

We wanted to inform you that we have sent the article to the MDPI editing service, but due to the holidays, they will not have it ready until next Monday (January 4th). In order not to delay the deadlines required by the editor, we announced the changes made.

We wish you a happy new year.

  • Reviewer 1 comment 1: I think a little editing of the discussion section would make it easier to read and understand. Specificially, shorter paragraphs focusing on one point /

We have made the suggested changes. You can see them throughout the discussion section. Thanks a lot.

  • Reviewer 1 comment 2: I don't think too many of us should be surprised by the findings. However, I think the importance of this paper would be raised by a brief discussion of your findings in relationship to other stressors. [Or tell us more about why this study was undertaken.] /  

We agree with your comment, and therefore we have added several paragraphs. You can see them from line 283 to 287.

  • Reviewer 1 comment 3: A larger number of subjects would have been better (always better). However in the age of Covid, this is very good. /

Thank you. We have mentioned it on line 274

Best wishes

Reviewer 2 Report

Review for manuscript #1050-366: “Empathy and Covid-19: Study in professionals and students of the social-health field in Ecuador”

This study investigates the four dimensions of empathy as represented in the factors of the Interpersonal Reactivity Index Questionnaire and their relationship in health-area personnel.  The manuscript is written well and reports the association between professionals and health students’ level of empathy.

The authors reported the introduction well. They covered conceptually the dimensions of perspective taking, fantasy, empathic concern, and empathy distress in the introduction.  One thing they could include is an example of a statement from the IRI Questionnaire that represents each dimension.  Since the IRI has also been utilized a lot in prior research, the authors could also report previous studies examining the IRI and its dimensions on related medical or health arenas, like trauma victims and psychological assistance, etc.  The establishment of these prior studies and their findings about the relationship between IRI or some of its dimensions on prosocial behavior or health related assistance would only contribute to the strength of their study and reported findings.

The authors also described the sample and the characteristics of the professional and student sample very well.  It might be better to present these demographic differences between the samples in a table.  The table of the resulting ttests conducted was extremely helpful and useful.  Maybe something like that, with reporting all means and standard deviations, could show all these characteristics that were measured among the subjects.

The most interesting outcome of this study as reported is the relationship between perspective-taking, empathy in general, and age.  It has been reported in prior studies that perspective taking becomes more of a marker of empathy as one ages.  This is made possible with a more expanded set of experiences in one’s health profession perhaps.  Empathy thus is increased since it may be perspective taking that play a more central role as one ages.

The many interesting results that the authors reported here from both the ttest and ANOVA analyses should be placed in a table or tables.  That would make it possible for the reader to see a wider, better, view of the findings.  These results are really interesting and it would help to organize the findings nicely in a table or tables.

The study obviously does not have IRI data from the professional and students during PRE-Covid times.  The authors have to address the possibility that the levels of empathy, distress, perspective-taking, and fantasy levels that they reported here may be reflective of the samples’ empathy pre-covid times.  Granted that the pandemic has increased the strain and stress in all health-related fields and personnel all over the world, we would still want to know by how much has Covid affected these dimensions of empathy.  Having data or measurements before the coronavirus showed up in the scene would have helped determine that computing and reporting difference scores, as opposed to only postest scores, may be more reflective of empathy levels today.

Author Response

Dear reviewer 2,

We want to give you our sincere thanks for helping to improve the paper. In the following points, we reply to your constructive remarks.

We wanted to inform you that we have sent the article to the MDPI editing service, but due to the holidays, they will not have it ready until next Monday (January 4th). In order not to delay the deadlines required by the editor, we announced the changes made.

We wish you a happy new year.

  • Reviewer 2 comment 1: The authors reported the introduction well. They covered conceptually the dimensions of perspective taking, fantasy, empathic concern, and empathy distress in the introduction.  One thing they could include is an example of a statement from the IRI Questionnaire that represents each dimension.

Thank you for your comment. We have added his suggestions. You can see them on line 85

  • Reviewer 2 comment 2 : Since the IRI has also been utilized a lot in prior research, the authors could also report previous studies examining the IRI and its dimensions on related medical or health arenas, like trauma victims and psychological assistance, etc.  The establishment of these prior studies and their findings about the relationship between IRI or some of its dimensions on prosocial behavior or health related assistance would only contribute to the strength of their study and reported findings.

Thank you for your suggestions. We have completed the information. You can see it between lines 36-50 and 97-99.

  • Reviewer 2 comment 3: The authors also described the sample and the characteristics of the professional and student sample very well.  It might be better to present these demographic differences between the samples in a table.  The table of the resulting ttests conducted was extremely helpful and useful.  Maybe something like that, with reporting all means and standard deviations, could show all these characteristics that were measured among the subjects. Thanks for your wise suggestion. /

We have added table 1. You can see it on line 140.

  • Reviewer 2 comment 4: The most interesting outcome of this study as reported is the relationship between perspective-taking, empathy in general, and age.  It has been reported in prior studies that perspective taking becomes more of a marker of empathy as one ages.  This is made possible with a more expanded set of experiences in one’s health profession perhaps.  Empathy thus is increased since it may be perspective taking that play a more central role as one ages. /

Thanks for your wise suggestion. We have delved more about this topic in the discussion (line 240)

  • Reviewer 2 comment 5: The many interesting results that the authors reported here from both the ttest and ANOVA analyses should be placed in a table or tables.  That would make it possible for the reader to see a wider, better, view of the findings.  These results are really interesting and it would help to organize the findings nicely in a table or tables. /

Thank you, we have incorporated tables 3 and 4.

  • Reviewer 2 comment 6: The study obviously does not have IRI data from the professional and students during PRE-Covid times.  The authors have to address the possibility that the levels of empathy, distress, perspective-taking, and fantasy levels that they reported here may be reflective of the samples’ empathy pre-covid times.  Granted that the pandemic has increased the strain and stress in all health-related fields and personnel all over the world, we would still want to know by how much has Covid affected these dimensions of empathy.  Having data or measurements before the coronavirus showed up in the scene would have helped determine that computing and reporting difference scores, as opposed to only postest scores, may be more reflective of empathy levels today /

Thanks, we have incorporated this limitation in line 269.

Best wishes.

Reviewer 3 Report

This paper is relevant in current times. 

Here I mention some aspects to strengthen the manuscript.

  • Line 30. Authors should avoid using phrases such as “as we understand” when referring to theories and concepts which have been defined scientifically at the international level. Instead, include the definitions of those concepts (Perspective Taking, Empathic Concern, Personal Distress, and Fantasy) as elaborated by the international scientific community, including citations.
  • In the methods section, when describing participants, please indicate if those professionals had received any kind of training on empathy, is that was controlled and how.
  • Discussion. Lines 183-188. Please discuss those findings with scientific literature in the field.
  • Analysis. The number of patients attended was an information gathered. Was not it included in the analysis to see its influence upon the measured dimensions? If not, why? If the influence of the ratio of patients per doctor on the variables measured was not examined, is not it a limitation of the study and, therefore, should not it be pointed out in the discussion?
  • Discussion. How does this study advance the state of the art on empathy and the health system? Please include some discussion on this.
  • Discussion. Please, indicate some practical implications of your results for health professionals during in times of health crises.

Author Response

Dear reviewer 3,

We want to give you our sincere thanks for helping to improve the paper. In the following points, we reply to your constructive remarks.

We wanted to inform you that we have sent the article to the MDPI editing service, but due to the holidays, they will not have it ready until next Monday (January 4th). In order not to delay the deadlines required by the editor, we announced the changes made.

We wish you a happy new year.

  • Reviewer 3, comment 1: Line 30. Authors should avoid using phrases such as “as we understand” when referring to theories and concepts which have been defined scientifically at the international level. Instead, include the definitions of those concepts (Perspective Taking, Empathic Concern, Personal Distress, and Fantasy) as elaborated by the international scientific community, including citations. /

Thank you, we have incorporated the changes in the first paragraph (line 24-32)

  • Reviewer 3, comment 2: In the methods section, when describing participants, please indicate if those professionals had received any kind of training on empathy, is that was controlled and how.

 Thank you. We have answered your comment on line 73.

  • Reviewer 3, comment 3: Discussion. Lines 183-188. Please discuss those findings with scientific literature in the field.

 Thank you. We have expanded the discussion (line 240-256)

  • Reviewer 3, comment 4: Analysis. The number of patients attended was an information gathered. Was not it included in the analysis to see its influence upon the measured dimensions? If not, why? If the influence of the ratio of patients per doctor on the variables measured was not examined, is not it a limitation of the study and, therefore, should not it be pointed out in the discussion? /

 We did not find significant data in the results. We have indicated it on the line 186.

  • Reviewer 3, comment 5: Discussion. How does this study advance the state of the art on empathy and the health system? Please include some discussion on this.

We have made a change in the discussion section. You can see the new paragraph from line 292 to 293.

  • Reviewer 3, comment 6: Discussion. Please, indicate some practical implications of your results for health professionals during in times of health crises.

Thank you. You will be able to see some of these practical implications from lines 277 to 282. 

Best wishes. 
